# Crystal structures of the mitochondrial deacylase Sirtuin 4 reveal isoform-specific acyl recognition and regulation features

Martin Pannek[1], Zeljko Simic[2], Matthew Fuszard[1], Marat Meleshin[2], Dante Rotili[3], Antonello Mai[3], Mike Schutkowski[2] & Clemens Steegborn[1]

Sirtuins are evolutionary conserved NAD[+]-dependent protein lysine deacylases. The seven human isoforms, Sirt1-7, regulate metabolism and stress responses and are considered therapeutic targets for aging-related diseases. Sirt4 locates to mitochondria and regulates fatty acid metabolism and apoptosis. In contrast to the mitochondrial deacetylase Sirt3 and desuccinylase Sirt5, no prominent deacylase activity and structural information are available for Sirt4. Here we describe acyl substrates and crystal structures for Sirt4. The enzyme shows isoform-specific acyl selectivity, with significant activity against hydroxymethylglutarylation. Crystal structures of Sirt4 from *Xenopus tropicalis* reveal a particular acyl binding site with an additional access channel, rationalizing its activities. The structures further identify a conserved, isoform-specific Sirt4 loop that folds into the active site to potentially regulate catalysis. Using these results, we further establish efficient Sirt4 activity assays, an unusual Sirt4 regulation by NADH, and Sirt4 effects of pharmacological modulators.

[1] Department of Biochemistry, University of Bayreuth, 95440 Bayreuth, Germany. [2] Department of Enzymology, Institute of Biochemistry and Biotechnology, Martin-Luther-University Halle-Wittenberg, 06108 Halle, Germany. [3] Department of Chemistry and Technologies of Drugs, Pasteur Institute Italy, Cenci-Bolognetti Foundation, Sapienza University of Rome, 00185 Rome, Italy. Correspondence and requests for materials should be addressed to C.S. (email: Clemens.Steegborn@uni-bayreuth.de)

Reversible acetylation of protein Lys side-chains is a post-translational modification in all domains of life. More than 7000 mammalian acetylation sites are known, and many of them regulate various target functions[1,2]. Among the protein Lys deacetylases, sirtuins form the evolutionary defined class III. They catalyze an unusual, NAD$^+$-dependent deacetylation reaction, coupling their activity to the metabolic state[3]. The seven mammalian sirtuin isoforms are primarily located in nucleus (Sirt1, 6, 7), cytosol (Sirt2), or mitochondria (Sirt3, 4, 5), and they regulate

processes from metabolism to stress responses[2,4]. Sirtuins have further been implicated in aging-related diseases, such as metabolic disorders and neurodegeneration, and are considered potential therapeutic targets[5,6].

Sirt4 acts as a metabolic regulator. It inhibits malonyl-CoA-decarboxylase (MCD), which represses fatty acid oxidation and promotes lipid anabolism[6,7], and it inhibits pancreatic glutamate dehydrogenase (GDH) to regulate insulin secretion[8,9]. Sirt4 further inhibits pyruvate dehydrogenase (PDH)[10] and stimulates

**Fig. 1** Sirt4 deacylation activities. **a** Chemical structures of CPS1 peptide and Lys acylations; from top: acetylation, butyrylation, DMS-ylation, HMG-ylation. For the complete set of acyl modifications see Supplementary Fig. 1a. **b** Sirt4-dependent deacylation of differently acylated CPS1 peptides. ($n = 2$; error bars: s.d.). **c** Sirt4 titrations with CPS1 substrate peptide carrying an acetyl, lipoyl, HMG, or DMS modification, respectively. ($n = 2$; error bars: s.d.). **d** Comparison of Sirt3, 4, and 5 deacylation activities against substrate peptide with acetyl, succinyl, DMS, or HMG modification, respectively. ($n = 2$; error bars: s.d.). **e** Intact protein mass spectrometry of HMG-ylated CypA (unmodified molecular weight 18,012 Da). **f** Sirt4-dependent deacylation reactions with increasing amounts of untreated and HMG-ylated CypA protein, respectively, as a substrate. ($n = 2$; error bars: s.d.). **g** Comparison of the acyl selectivities of Sirt4 from human (hSirt4), clawed frog (xSirt4), and zebrafish (zSirt4) using CPS1 peptide substrates featuring an acetyl, succinyl, DMS, itaconyl, HMG, or lipoyl modification, respectively. ($n = 2$; error bars: s.d.)

mitochondrial ATP production[11]. Due to these effects on energy metabolism, Sirt4 is considered a therapeutic target for metabolic dysfunctions[6,7]. Furthermore, Sirt4 displays tumor suppressor activity through downregulation of glutamine metabolism and has been implicated in several cancer types[6,12].

Sirtuins are increasingly recognized as deacylases with isoform-specific acyl selectivities, catalyzing removal of acylations emerging as posttranslational protein modifications, such as succinylation or crotonylation[13–15]. While Sirt1–3 are strong deacetylases, Sirt5 shows low-deacetylation activity and acts primarily as a desuccinylase and deglutarylase[13,15,16], and Sirt6 deacetylates histones but displays more prominent demyristoylation activity[17]. Similarly, Sirt4 features weak deacetylation activity, which appears to regulate MCD[7], but for most of its functional effects the catalyzed target modification appears to differ or is unknown[6]. It can inhibit PDH through delipoylation[10], but the catalytic efficiency for this reaction appears much lower than for other primary sirtuin activities[13]. ADP-ribosyltransferase activity was also described for Sirt4 and Sirt6, but it is also inefficient and appears to constitute a side-activity[8,18], so that a prominent Sirt4 enzyme activity remains to be identified.

Sirtuins share a conserved catalytic core of ~275 amino acids[19]. Isoform-specific N-terminal and C-terminal domains contribute to regulation and cellular localization[6,20]. In Sirt4, the core has no C-terminal appendage and only a short, ~28 residue N-terminal extension that serves as mitochondrial localization sequence[6,9]. The sirtuin core comprises a Rossmann-fold subdomain and a smaller $Zn^{2+}$-binding module[21,22]. $NAD^+$ and the acylated substrate polypeptide are bound, with moderate sequence selectivity, to a cleft between the subdomains, accompanied by closure movements of the subdomains and a flexible "cofactor-binding loop"[1,21–24]. The ribose then releases nicotinamide (NAM), and via an 1′-O-alkylimidate and a bicyclic intermediate the products, deacetylated polypeptide and 2′-O-acyl-ADP-ribose, are formed[3,19]. This mechanism was deduced from biochemical studies and crystal structures including human Sirt1, 2, 3, 5, and 6[3,19,21,22,25,26], and it applies to all sirtuin-dependent deacylations. The isoform differences in preferred substrate acyls are caused by binding of the acyl moiety to an active site channel with isoform-specific features. For Sirt4, however, a lack of structural and enzymatic data hampers insights in Sirt4 acyl specificity and regulation.

Here we report crystal structure and enzymatic characterization of Sirt4. We identify an evolutionary conserved, Sirt4-specific acyl selectivity and dehydroxymethylglutarylation (de-HMG-ylation) as a potential physiological activity. A structure of Sirt4 from *X. tropicalis* reveals an unusual acyl binding site and a Sirt4-specific, potentially regulatory loop. Using these insights, we analyze and rationalize Sirt4 modulator effects and identify a Sirt4 regulation by NADH.

## Results

**Sirt4 shows an isoform-specific acyl preference.** The seven mammalian sirtuins vary in their sequence and acyl preferences[16,19]. The Sirt4 deacylation activities reported so far, deacetylation and delipoylation, were weak, with orders of magnitude lower $k_{cat}/K_M$ values as for other sirtuin/substrate acyl pairs[7,10,15]. Testing ~6800 mammalian acetylation site sequences yielded no dramatic activity improvements[1]. We therefore asked whether other acyl modifications would yield deacylation efficiencies expected for a physiologically dominant Sirt4 activity. Testing Sirt4 against an acyl library of CPS1 (carbamoyl phosphate synthetase 1)-Lys527 peptides in a coupled enzymatic assay monitoring NAM release from $NAD^+$ [27] indeed revealed a

**Table 1 Kinetic parameters for Sirt4 and acyl substrates**

|  | $k_{cat}$ ($10^{-3}$ s$^{-1}$) | $K_M$ (μM) | $k_{cat}/K_M$ (M$^{-1}$ s$^{-1}$) |
|---|---|---|---|
| Acetyl-CPS1 | 8.7 ± 0.7 | 2341 ± 270 | 3.7 ± 0.7 |
| Acetyl-DLAT[a] | ND[a] | ND (>2500)[a] | 0.2 ± 0.0 (estimated)[a] |
| Lipoyl-CPS1 | 1.7 ± 0.3 | 10.1 ± 13.6 | 170 ± 230 |
| Lipoyl-DLAT[a] | 1.8 ± 0.1[a] | 239 ± 51[a] | 7.7 ± 1.3[a] |
| DMS-CPS1 | 7.3 ± 0.1 | 17.7 ± 1.5 | 412 ± 41 |
| HMG-CPS1[b] | 5.3 ± 0.1 | 9.7 ± 1.0 | 546 ± 67 |

[a]Values from[10]
[b]Not corrected for HMG-stimulated $NAD^+$ hydrolysis

particular specificity profile (Fig. 1a, b, Supplementary Fig. 1a). Consistent with previous reports[1,7,10], Sirt4 showed low-deacetylation activity but higher activity against lipoylated and biotinylated substrate (Fig. 1b). Further increased activity was obtained with butyryl and octanoyl substrate, but the highest activity—eightfold stronger than deacetylation—was observed with a 3,3-dimethylsuccinyl (DMS) substrate.

Comparing DMS to acyl moieties physiologically occurring as activated CoA-thioesters, and thus potentially modifying protein Lys side chains[28,29], revealed the 3-hydroxy-3-methylglutaryl (HMG) group as most closely related. Testing HMG-modified CPS1-Lys527 peptide indeed yielded Sirt4 activity similarly to DMS-CPS1 substrate (Fig. 1b), approximately threefold higher than for acetyl peptide and with the expected $NAD^+$ dependency (Supplementary Fig. 1b). To analyze the mechanistic basis of these differences, we compared through Michaelis-Menten kinetics the improved substrates, HMG- and DMS-CPS1, with lipoyl- and acetyl-CPS1 (Fig. 1c, Table 1). Consistent with previous data[10,16], catalytic efficiency $k_{cat}/K_M$ for acetyl-CPS1 was low (3.7 ± 0.7 M$^{-1}$ s$^{-1}$; Table 1). Sirt4 activity was strongly increased for DMS-CPS1 (412 ± 41 M$^{-1}$ s$^{-1}$) and HMG-CPS1 (546 ± 67 M$^{-1}$ s$^{-1}$). Strikingly, the preferred acyls showed comparable turnover rates to acetyl substrate but increased apparent affinities (two orders of magnitude lower $K_M$; Fig. 1c, Table 1). Interestingly, lipoyl-CPS1 yielded an only slightly lower $k_{cat}/K_M$ (170 ± 230 M$^{-1}$ s$^{-1}$), two orders of magnitude higher than a previously published value[10], due to a better $K_M$ in our study, comparable to those for the DMS/HMG modifications (Table 1). These results indicate that lipoylated substrates bind much better to Sirt4 than so far known, but DMS/HMG still yield 3–5-fold higher efficiencies due to faster turnover. Importantly, the catalytic efficiencies with the DMS/HMG substrates are close to those of robust sirtuin activities, such as Sirt2-dependent deacetylation (e.g., 1400 M$^{-1}$ s$^{-1}$ for histone H3-K27[30]). Sirt4 thus features significant deacylation activity and appears to discriminate acyl substrates mainly via their apparent binding affinity.

Sirtuin-dependent NAM release from $NAD^+$ is normally coupled to deacylation. However, when we analyzed Sirt4-dependent turnover of HMG-CPS1 through MS detection of substrate and product peptide parallel to monitoring $NAD^+$ hydrolysis, HMG-CPS1 deacylation corresponded only to ~40 % of the $NAD^+$ turnover (Supplementary Fig. 1c). For Sirt4-dependent conversion of acetyl-CPS1, in contrast, no discrepancy to $NAD^+$ hydrolysis was observed (Supplementary Fig. 1c). Sirt4 thus shows significant $NAD^+$-dependent de-HMG-ylation activity, but it catalyzed even slightly better HMG-stimulated $NAD^+$ glycohydrolysis as an unusual, deacylation independent sirtuin activity.

We next compared the selectivities of the mitochondrial sirtuins Sirt3, 4, and 5 against CPS1 peptides carrying HMG-modifications and DMS-modifications or the generic Sirt3 and

**Table 2 Data collection and refinement statistics**

|  | xSirt4/ADPr | xSirt4/thioacetyl-ADPr | zSirt5/HMG-CPS1 |
|---|---|---|---|
| Space group | C222$_1$ | C222$_1$ | P6$_5$22 |
| Unit cell constants | $a = 69.4$ Å, $b = 74.7$ Å, $c = 109.7$ Å | $a = 69.0$ Å, $b = 74.9$ Å, $c = 109.6$ Å | $a = b = 87.5$ Å, $c = 316.9$ Å |
| Resolution[a] | 20.00–1.58 Å    (1.62–1.58 Å) | 20.00–1.80 Å    (1.85–1.80 Å) | 50.00–3.10 Å    (3.20–3.10 Å) |
| Unique reflections | 39,280 (2868) | 26,585 (1932) | 13,919 (1214) |
| Multiplicity | 5.1 (5.2) | 6.8 (7.1) | 10.3 (10.8) |
| Completeness | 99.8% (99.9%) | 99.7% (99.9%) | 99.9% (99.9%) |
| $R_{meas}$ | 3.6% (85.3%) | 5.5% (104.4%) | 24.8% (151.2%) |
| $CC1/2$ (%) | 100.0 (70.4) | 99.9 (70.4) | 99.4 (59.8) |
| $I/\sigma I$ | 23.7 (2.1) | 18.3 (2.1) | 10.2 (1.6) |
| Protein atoms | 2285 | 2177 | 4138 |
| Ligand atoms | 37 | 40 | 173 |
| Solvent atoms | 285 | 179 | 68 |
| Resolution | 19.73–1.58 Å    (1.62 Å – 1.58 Å) | 19.74–1.80 Å    (1.85–1.80 Å) | 48.67–3.10 Å    (3.18–3.10 Å) |
| $R_{cryst}/R_{free}$[bc] | 15.2%/18.6% | 15.5%/20.8% | 19.6%/26.6% |
| *Average B-factors* |  |  |  |
|  Protein | 30.9 | 39.4 | 73.4 |
|  Ligands | 21.8 | 30.4 | 74.3 |
|  Solvent | 41.6 | 46.3 | 48.5 |
| RMSD bond-lengths | 0.03 | 0.03 | 0.01 |
| RMSD bond-angles | 2.6 | 2.4 | 1.5 |

[a]Values in parentheses refer to outermost shell

[b] $R_{cryst} = \dfrac{\sum ||F_{obs}| - k|F_{calc}||}{\sum |F_{obs}|}$. $|F_{obs}|$ is the observed and $|F_{calc}|$ the calculated structure factor amplitude

[c]$R_{free}$ was calculated from 5% of reflections omitted from refinement

5 substrate modifications acetylation (Sirt3) and succinylation (Sirt5). Sirt4 showed the previous preference for DMS and HMG modifications, and very low-activity against acetylated or succinylated peptide (Fig. 1d). Sirt3, in contrast, featured pronounced selectivity for the acetyl modification, minor activity against DMS substrate, and no turnover with HMG and succinyl substrate (Fig. 1d). The other way round, Sirt5 showed no activity against acetyl substrate but high activity against succinyl peptide, and lower but still significant activity against DMS- and HMG-CPS1 consistent with reported parameters for HMG substrate ($K_M = 8$ μM,  $k_{cat}/K_M = 500$ M$^{-1}$ s$^{-1}$)[15]. Sirt5 activity against HMG-CPS1 thus exceeds that of Sirt4 mainly due to faster turnover, and this Sirt5 activity appears even higher with other peptide sequences, but it is still 60–80% lower than the enzyme's desuccinylase activity (Fig. 1d). In summary, Sirt4 shows a particular acyl preference profile: it shares de-HMG-ylation but no desuccinylation activity with Sirt5 and shows no activity overlap with Sirt3, and it features unusual activity against DMS substrate and for HMG-stimulated NAD$^+$ hydrolysis. This distinct profile suggests that there might be additional protein Lys acylations that are specifically removed by Sirt4.

**HMG-CoA acylates and Sirt4 de-HMG-ylates proteins**. Acetyl transferases employ acetyl-CoA for modifying proteins. However, acetyl-CoA can also non-enzymatically modify proteins, and some other acyl-CoA, such as succinyl-CoA, do so even more efficiently, likely causing the emerging variety of protein acylations[14,28,29,31]. We thus analyzed whether HMG-CoA efficiently acylates peptides and proteins. HMG-CoA titrations revealed CPS1 peptide acylation with a bimolecular rate constant of $(4.0 \pm 0.8) \times 10^{-10}$ μM$^{-1}$ s$^{-1}$, more than twice as fast as with acetyl-CoA ($k = (1.5 \pm 0.4) \times 10^{-10}$ μM$^{-1}$ s$^{-1}$)[29], in agreement with a recent study reporting more efficient protein acylation by glutaryl- and HMG-CoA as compared to acetyl-CoA[28]. Incubating recombinant Cyclophilin A (CypA) as a model protein with HMG-CoA resulted in 1–7 HMG-modifications as detected by intact protein mass spectrometry (MS; Fig. 1e) and confirmed by MS/MS

analysis of tryptic peptides, consistent with the protein's seven known Lys acetylation sites (Uniprot entry P62937).

To analyze whether Sirt4 is able to remove HMG modifications not only from peptides but also from HMG-ylated protein, we tried to de-HMG-ylate CypA. Modified and unmodified CypA was incubated with Sirt4, in the presence and absence of NAD$^+$, and the deacylation was monitored in the coupled enzymatic assay[27]. Unmodified CypA as substrate did not yield a significant deacylation signal, whereas HMG-CypA substrate resulted in a strong, substrate concentration dependent signal (Fig. 1f) that showed the expected dependency on the co-substrate NAD$^+$ (Supplementary Fig. 1d). To confirm that NAD$^+$-dependent CypA de-HMG-ylation causes or significantly contributes to the NAM release monitored in this assay, we also analyzed the reaction by intact protein mass spectrometry. Incubation with Sirt4 indeed caused a shift toward CypA species carrying fewer HMG-ylations (Supplementary Fig. 1e), confirming the deacylation. We thus conclude that HMG-CoA is reactive toward proteins and that Sirt4 can de-HMG-ylate the modified proteins, consistent with recently published work that furthermore confirmed the physiological occurrence of protein HMG-ylation and Sirt4-dependent de-HMG-ylation[28,32].

**Crystal structure of Sirt4**. For insights in the molecular basis of Sirt4's substrate preference and other isoform-specific features, we solved a Sirt4 crystal structure. Trials to crystallize our human Sirt4 protein construct (residues 25–314), which comprises the catalytic core with native C-terminus but lacks 24 residues of the N-terminal mitochondrial localization sequence (MLS; residues 1–28[9]), were not successful. Therefore, we used the Sirt4 orthologues from *Xenopus tropicalis* (clawed frog; xSirt4) and *Danio rerio* (zebrafish; zSirt4) as model systems. They show significant sequence deviations only in the ~30 N-terminal residues (Supplementary Fig. 2a), consistent with their function as MLS, which tend to show low-sequence conservation[33]. Within the catalytic core (hSirt4: 32–312; xSirt4: 31–314; zSirt4: 28–310), however, the sequence identity with hSirt4 is 67% (xSirt4; similarity 81%) and

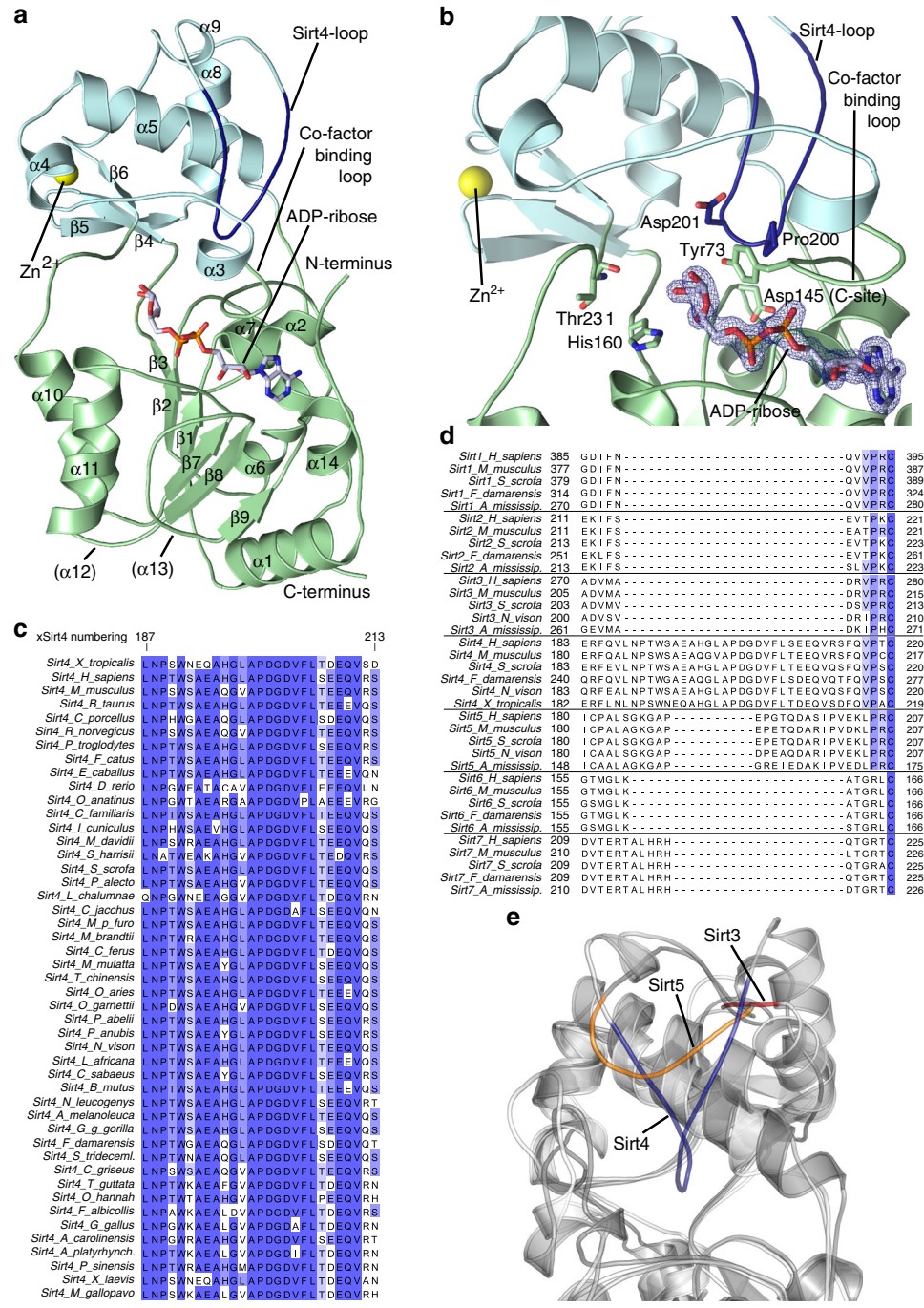

**Fig. 2** Crystal structure of xSirt4. **a** Overall structure of the xSirt4/ADPr complex, with Rossmann-fold domain (green), Zn²⁺-binding domain (cyan), and a Sirt4-specific loop (blue) indicated. ADPr is shown as sticks, colored according to atom type. Secondary structure elements are numbered equivalent to other sirtuins, elements missing in xSirt4 are indicated by brackets. **b** xSirt4 active site, with conserved sirtuin catalytic residues and key residues of the Sirt4-loop shown as sticks. ADPr sticks are colored according to atom type and overlaid with $2F_o–F_c$ electron density ($1\sigma$). **c** Alignment of the Sirt4-loop region in Sirt4 sequences from various chordates (Full alignment: Supplementary Fig. 2c). **d** Section of a structure-based alignment of Sirt1–6, extended by chordate Sirt1–7 sequences, showing the isoform differences in the Sirt4-loop region (Full alignment: Supplementary Fig. 2d). **e** Overlay of xSirt4 (gray, blue) with Sirt3 (light gray, red; PDB ID 4BVH) and 5 (dark gray, orange; 4G1C) showing the extended Sirt4-loop, the shorter Sirt5 surface loop, and the short turn in Sirt1–3 (represented by Sirt3)

65% (zSirt4; similarity 78%), respectively (Supplementary Fig. 2a). Consistent with the high sequence conservation, xSirt4 and zSirt4 showed the same acyl preferences as hSirt4 (Fig. 1g), confirming that this acyl selectivity is an evolutionary conserved Sirt4 feature and that xSirt4 and zSirt4 are suitable models for the mammalian enzyme.

xSirt4 yielded well diffracting crystals in presence of ADP-ribose (ADPr). The xSirt4/ADPr structure was solved through Patterson searches with Sir2Af1 (PDB entry 4TWI) and refined at 1.58 Å resolution to $R_{cryst}/R_{free}$ values of 15.2%/18.6% (Table 2). The xSirt4 overall structure shows the typical sirtuin architecture with Rossmann-fold domain and smaller Zn²⁺-binding module,

**Table 3 Kinetic parameters for xSirt4 wild-type and variants[a]**

| xSirt4 variant | Acetyl-CPS1 $K_M$ (µM) | Acetyl-CPS1 $v_{max}$ ($10^{-3}$ s$^{-1}$) | HMG-CPS1 $K_M$ (µM) | HMG-CPS1 $v_{max}$ ($10^{-3}$ s$^{-1}$) |
|---|---|---|---|---|
| Wild-type | 663 ± 69 | 12.9 ± 0.6 | 6.8 ± 0.9 | 15.9 ± 0.4 |
| Delta 196–205 + GSS | 541 ± 51 | 10.7 ± 0.4 | 9.7 ± 0.7 | 18.4 ± 0.3 |
| Delta 196–205 | 1599 ± 563 | 15.7 ± 3.4 | 8.5 ± 0.6 | 17.2 ± 0.2 |
| Delta 198–203 | 1104 ± 110 | 17.1 ± 0.9 | 10.1 ± 0.7 | 19.9 ± 0.3 |
| D201A | 707 ± 127 | 9.1 ± 0.8 | 11.3 ± 1.4 | 13.0 ± 0.3 |
| D203A | 925 ± 136 | 10.5 ± 0.8 | 9.0 ± 1.1 | 14.1 ± 0.3 |
| Y73F | 640 ± 71 | 8.8 ± 0.4 | 9.0 ± 1.0 | 24.1 ± 0.5 |
| R101A | ND | ND | 23.2 ± 1.3 | 13.3 ± 0.2 |
| Y104F | 643 ± 40 | 13.3 ± 0.4 | 15.3 ± 1.3 | 15.2 ± 0.3 |
| R107A | 1097 ± 159 | 11.0 ± 0.9 | 17.2 ± 1.8 | 14.2 ± 0.3 |
| Y104F R107A | 918 ± 72 | 18.5 ± 0.8 | 22.6 ± 2.6 | 15.7 ± 0.4 |
| N108A | 1104 ± 254 | 14.8 ± 1.9 | 19.3 ± 2.8 | 12.2 ± 0.4 |

[a]Not corrected for HMG-stimulated NAD$^+$ hydrolysis

with the active-site located in between them and harboring the nucleotide that is well defined by electron density (Fig. 2a, b). The six-stranded β-sheet of the Rossmann-fold domain provides a docking patch for the nucleotide, orienting the reacting ribose close to the conserved catalytic sirtuin residue His160 (Fig. 2b; numbering refers to xSirt4 if not stated otherwise). Due to the occupied nucleotide binding site, the cofactor-binding loop between α2 and α3 is in the "closed" conformation[23], positioning the conserved Phe/Tyr (Tyr73) on top of the ADPr ribose. Thermal denaturation shift experiments showed significant Sirt4 stabilization by NAD$^+$ or ADPr, while substrate peptide and NAM had no pronounced effects (Supplementary Fig. 2b), indicating that the nucleotide-induced closed conformation stabilizes the protein and facilitates crystallization.

**A Sirt4-specific loop and additional active site entry.** Comparing the Sirt4 structure to other sirtuin isoforms reveals as most striking difference an extended, ~12 residues Sirt4 loop in the Zn$^{2+}$-binding module, between α8 and α9 (residues 195–206; Fig. 2a). The loop is oriented deep into the catalytic core and contributes to the active site lining (Fig. 2b). A sequence comparison of all higher eukaryotic Sirt4 orthologs in the UNIPROT database shows a high overall sequence homology (89%), and in particular a strict conservation of the presence and even sequence of this loop (G(L/V)APDGDVFL(T/S)(D/E)EQ motif; Fig. 2c, Supplementary Fig. 2c). Structure-based comparison of human Sirt1–6, with human Sirt7 and Sirt1–7 sequences from other chordates added based on homology, shows that the loop is absent in all other sirtuin isoforms (Fig. 2d, Supplementary Fig. 2d), and we therefore refer to it as "Sirt4-loop". The structure-based comparison shows that the Sirt4-loop is extended compared to a much smaller Sirt5 surface loop, and to a short turn in other isoforms, and only the Sirt4-loop is thus able to reach the active site (Fig. 2b, e). Removing the loop (Δ189–214 and Δ192–212) yielded insoluble protein, but deleting the loop's core (Δ196–205, Δ198–203) or replacing it with a GSS linker (Δ196–205 + GSS) resulted in soluble and active protein, which shows that the extended loop is not essential for Sirt4's structural integrity. The variants tended to show higher $K_M$ values for the peptide substrate and at the same time slightly increased turnover (Table 3), which indicates that the loop contributes to substrate binding and restricts catalytically relevant active site dynamics (see also below). In a second xSirt4 structure, solved in complex with the product analog 2′-thioacetyl-ADPr, the Sirt4-loop was not defined by electron density (Table 2, Supplementary Fig. 2e), indicating that it is either flexible or can assume several conformations. Both xSirt4 complexes were solved from the same

crystal form, with residues 191–207 not participating in crystal contacts, excluding that crystal packing causes the loop differences. The Sirt4/2′-thioacetyl-ADPr structure might thus suggest that the loop is released from the active site during product formation, but functions and triggers of Sirt4-loop conformations remain to be studied in more detail.

Another unusual feature of the Sirt4 catalytic core is a channel that branches off from the acyl-Lys binding tunnel and leads to the protein surface (Fig. 3a, b). It is lined by residues from α4 and the preceding loop (86-ArgArgProIle, Glu93), α5 (Ala100, 103-ArgTyr, Arg107), and Ala199/Asp203 from the Sirt4-loop (Fig. 3a). They are mostly conserved in Sirt4, but differ in other isoforms (Fig. 3b). In Sirt6, the Sirt4 channel area is blocked by its isoform-specific N-terminus, but lack of α4, part of α5, and both loops results in a differently oriented, wide cleft that accommodates myristoyl substrates and activators[34]. Sirt1–3 and Sirt5 contain structure elements sized comparable to Sirt4, except for the missing Sirt4-loop, with sequences similar to each other but differing from Sirt4 (Fig. 3b, Supplementary Fig. 2d). The resulting fold in Sirt1–3 and Sirt5 comprises a more wiggled α3/α4 loop folded against α5 and blocking the Sirt4 channel area (Supplementary Fig. 3a). The Sirt4 channel has a small positively charged patch at the outer entrance but is generally rather hydrophobic, and we speculate that it contributes to accommodation of longer substrate acyls. Modeling a Sirt4/lipoyl-Lys complex indeed places the distal lipoyl end into the bottom of the channel (Fig. 3a), supporting an acyl binding function and rationalizing Sirt4's delipoylation activity. Interestingly, the channel could also accommodate a lipoyl group entering from outside as a substrate anchor, in a scenario related to SirTM, which catalyzes ADP-ribosylation of targets only after their lipoylation[35]. Furthermore, the channel might serve as a binding site for regulatory metabolites, similar to Sirt6 activation by free fatty acids[16]. Testing the effects of fatty acids, lipoic acid, and the ketone bodies beta-hydroxybutyrate and acetoacetate indeed revealed that free lipoic acid inhibits Sirt4's deacetylation and de-HMG-ylation activity (Supplementary Fig. 3b). However, the role of the Sirt4 channel in this and/or other regulation mechanisms remains to be studied in detail.

Using our structure-based alignment of Sirt4 and other structurally characterized sirtuins, extended by chordate Sirt1–7 sequences (Fig. 2d, Supplementary Fig. 2d), we analyzed phylogenetic relationships. The phylogenetic tree (based on 208 sequences; Fig. 3c) confirms some classifications based purely on sequence information[36] but also reveals modifications. Sirt4 forms a separate class (class II in ref. [36]) that is almost equally distant to classes I (Sirt1–3), III (Sirt5), and IV (Sirt6/7), despite its partial deacylation activity overlap with Sirt5. Prokaryotic

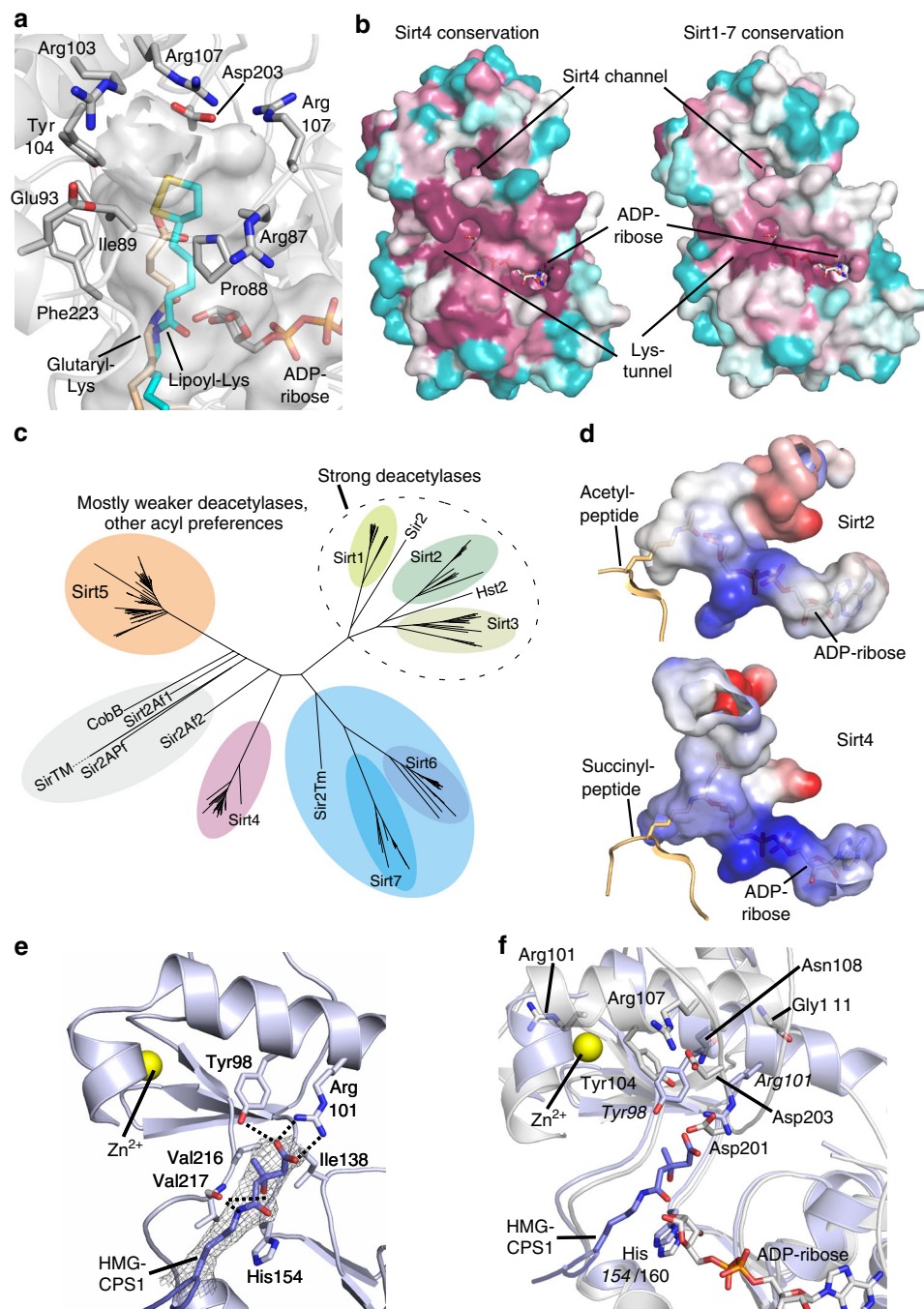

**Fig. 3** Sirt4 structural features and phylogeny. **a** xSirt4 active site with the additional Sirt4 channel to the acyl pocket shown as transparent surface. ADPr and residues forming the channel are shown as sticks colored according to atom type. Glutaryl-Lys (beige) from an overlaid Sirt5 complex (PDB ID 4UTR) indicates the conventional acyl pocket, and the modeled lipoyl-Lys (cyan) the bottom of the Sirt4 channel. **b** xSirt4 surface colored according to sequence conservation within Sirt4 isoforms (left; from higher eukaryotic Sirt4 in UniProt) and within the complete Sirtuin family Sirt1–7 (right; from chordate Sirt1–7 in UniProt). Purple indicates high conservation, cyan high variability. **c** Phylogenetic tree generated from a structure-based sirtuin alignment, extended by aligning 195 chordate sirtuin sequences (see Supplementary Fig. 2c for a core alignment). **d** Active site inner surface of Sirt4 and Sirt2 (PDB ID 5D7O; see Supplementary Fig. 3d for all isoforms) colored according to electrostatic potential (red: −15 to blue:+15 $k_BT/e$). The succinyl- and acetyl-peptide are from overlays (PDB IDs 3RIY and 3GLR, respectively). **e** Crystal structure of zSirt5 in complex with HMG-CPS1 substrate peptide. Ligand and interacting residues are shown as sticks, and 2$F_o$–$F_c$ electron density for the peptide is contoured at 1.0$\sigma$. Dotted lines indicate hydrogen bonds. **f** Active site overlay of xSirt4/ADPr (gray) and zSirt5/HMG-CPS1 (blue). Catalytic His and residues analyzed for acyl recognition contributions are shown as sticks and labeled (italics: zSirt5)

sirtuins form a cluster in the Sirt5 branch, separate from chordate Sirt5 and almost equidistant to Sirt4. The Sirt6/7 branch also comprises a prokaryotic member, Sir2Tm. The three subclusters for the strong deacetylases Sirt1, 2, and 3 are clearly separated from the other branches but contain yeast enzymes, indicating

that a deacetylase developed still early during evolution and diversified further, possibly reflecting the prominent role of acetylation among the protein acylations. All other branches appear to constitute subfamilies with mostly weak deacetylases, which developed different acyl preference profiles for each cluster

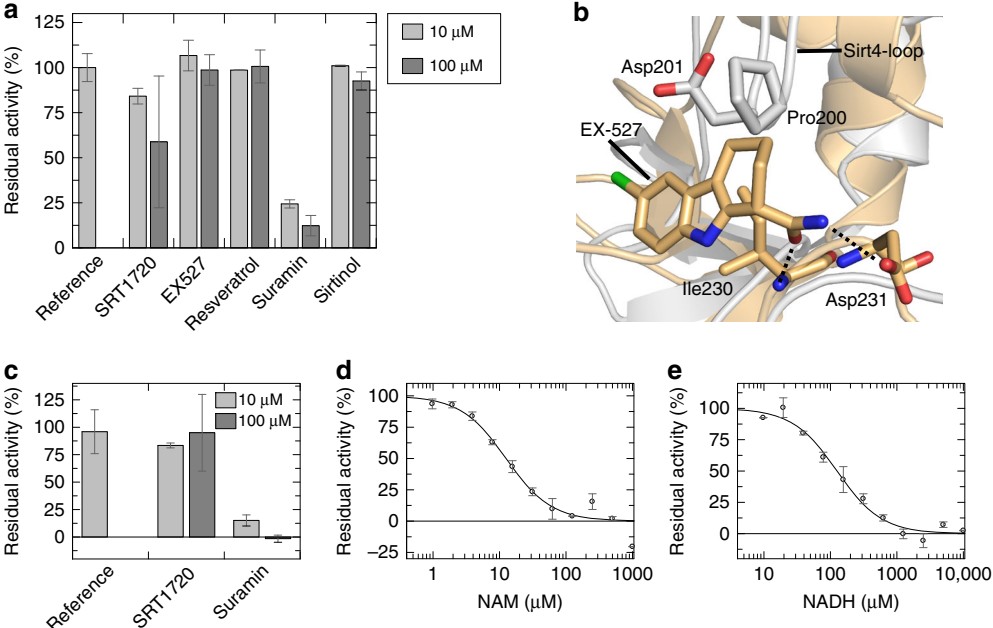

**Fig. 4** Sirt4 modulation by physiological metabolites and pharmacological compounds. **a** Effects of known sirtuin modulators on Sirt4 de-HMG-ylation activity. SRT1720 caused effects in controls, indicating incompatibility with the coupled enzymatic assay. ($n = 2$; error bars: s.d.). **b** C-site regions of xSirt4 (gray) and Sirt3/Ex527 (gold; PDB ID 4BVB), showing that Ex-527 would clash with the Sirt4-loop residue Pro200. Dotted lines: conserved hydrogen bonds for carbamide recognition. **c** Effects of SRT1720 and suramin on Sirt4 activity in a HMG-FdL assay. ($n = 2$; error bars: s.d.). **d** NAM titration of Sirt4 activity in a HMG-FdL assay. ($n = 2$; error bars: s.d.). **e** NADH titration of Sirt4 activity in a HMG-FdL assay (see Supplementary Fig. 4b for a titration in a MS assay). ($n = 2$; error bars: s.d.)

that remained conserved in later stages of evolution as indicated by our Sirt4 ortholog comparison. These major branches apparently separated early during evolution, consistent with a variety of acylations occurring non-enzymatically[28,29,31] and thus likely being evolutionary early posttranslational modifications that existed before the protein modifying enzymes emerged.

**Sirt4 nucleotide and acyl binding sites**. All sirtuins share NAD$^+$ as an essential cosubstrate, but the Sirt4 nucleotide binding site differs from those of other isoforms. The Sirt4-loop tightens the entry of the C-site, where the conserved Asp145 recognizes the NAM moiety of NAD$^+$[25], and its tip around Pro200 will have to rearrange slightly to allow productive NAD$^+$ binding (Supplementary Fig. 3c), consistent with a switch function. However, NAD$^+$ titrations revealed similar apparent NAD$^+$ affinities for wild-type Sirt4 ($K_M = 70 \pm 14\,\mu M$ for xSirt4; $62 \pm 14\,\mu M$ for hSirt4) and a Sirt4 loop deletion variant ($K_M = 66 \pm 9\,\mu M$; Supplementary Fig. 3d), suggesting that a regulatory factor stabilizing the inactive conformation might be missing, consistent with the loop variability observed in our structures (see above). Positive electrostatic potential in the nucleotide site supports binding of the negatively charged ligand in all isoforms, but in Sirt4 the positive potential is more pronounced than in most isoforms (Fig. 3d, Supplementary Fig. 3e), rationalizing its low $K_M$(NAD$^+$) within the 14–600 μM range reported for sirtuins[15,37,38]. Sirt4's cosubstrate site might also facilitate binding of NADH, compared to NAD$^+$, due to its lacking positive charge at the nicotinamide, and we indeed find a distinctive sensitivity of this sirtuin isoform to NADH (see below). In the peptide binding cleft, Sirt4 also features positive potential in regions accommodating the substrate sequence immediately N-terminal and C-terminal from the substrate acyl-Lys (Supplementary Fig. 3f). Consistently, peptide array studies with its weaker deacetylase activity had shown a

Sirt4 substrate sequence preference for polar residues around the acetyl-Lys, in particular with negative charges in positions +1/2 and −3/4[1].

In the active site, Sirt4 comprises the conserved sirtuin catalytic residues His160 and the Leu/Thr/Val (Thr231) whose backbone oxygen orients the substrate through a hydrogen bond to the acyl-Lys ε-amide (Fig. 2b)[15]. The cofactor-binding loop in "closed" conformation[23] positions the conserved Phe/Tyr (Tyr73) on top of the ribose, shielding this reacting group. Mutating Tyr73 to the Phe found in most sirtuins increased $k_{cat}$ (Table 3), confirming its catalytic relevance and indicating that its hydroxyl group might contribute to an auto-inhibitory mechanism. In the region accommodating a distal carboxyl group in short and medium long dicarboxylates, such as HMG, mildly positive electrostatic potential rationalizes the affinity of such substrates (Fig. 3d). Attempts to solve a Sirt4 structure in complex with substrate peptide to study acyl recognition details failed, possibly due to the lacking stabilization by this ligand (Supplementary Fig. 2b), but we were able to solve a complex of zebrafish Sirt5 (zSirt5) with HMG-CPS1 peptide. The structure, refined at 3.1 Å resolution to $R_{cryst}/R_{free}$ values of 19.5%/26.6% (Table 2, Fig. 3e), reveals a similar substrate conformation as in zSirt5/glutaryl-CPS1. The HMG length is reduced through a helical conformation, albeit with opposite handedness compared to glutaryl-CPS1, to position the 3-hydroxy and 3-methyl moieties toward Ile138/His154 and Tyr98/Val216, respectively (Fig. 3e, Supplementary Fig. 3g). The distal carboxylate interacts with Tyr98 and Arg101, which also recognize the carboxylates of succinyl and glutaryl substrates[15]. Interestingly, Sirt4 also features such a Tyr-X-X-Arg motif in α5. The Sirt4 motif is shifted one helix turn, however, with Sirt4-Tyr104 oriented back to the region occupied by Sirt5-Tyr98, but Sirt4-Arg107 pointing away from the acyl channel (Fig. 3f). Sirt4 residues in the positions of Sirt5-Tyr98/Arg101 would be Asn108 and Gly111. Due to a slightly different conformation, however,

Gly111 is shifted and the area of Sirt5-Arg101 is occupied by the Sirt4-loop residue Asp201 (Fig. 3f), which would result in close contacts between the carboxylates of Asp201 and an HMG substrate. To test the roles of these residues, we analyzed deacetylation and de-HMG-ylation kinetics of xSirt4 Ala variants (Table 3). Exchanging Asp201 or Asp203 caused only a moderate decrease in apparent HMG-peptide affinity, consistent with Sirt4-loop deletion effects. They thus appear not directly involved in substrate binding but to rearrange dynamically, as expected from the repellent effect Asp201 would have on substrate carboxylates and in line with our crystallographic results. Consistently, testing β-Ala-, γ-aminobutyryl-, and 6-aminocaproyl-CPS1 substrates (Supplementary Fig. 1a), which have positively charged end groups that could favorably interact with Asp201 in this position, yielded no Sirt4-dependent deacylation (Supplementary Fig. 3h). Replacing Tyr104 (to Phe) or Arg107 of the Sirt4 Tyr-X-X-Arg motif had a stronger, approximately twofold effect on HMG-peptide $K_M$ values, and a Tyr104Phe/Arg107Ala double mutant even showed an additive effect (Table 3). They thus seem to contribute—directly or indirectly—to substrate binding despite Arg107's orientation away from the active site. In fact, replacing the neighboring Asn108, which points toward the acyl channel, caused a significant, approximately threefold decrease in apparent HMG-substrate affinity, suggesting a contribution to acyl binding and possibly an indirect role for Arg107, via its salt bridge to the Sirt4-loop residue Asp203 (Fig. 3f). Interestingly, mutating Arg101 from a 101-ArgArgArg-103 motif on the opposite side of Tyr104 caused a strong, approximately fourfold decrease in apparent HMG-peptide affinity (Table 3), indicating an important role in substrate recognition despite its distance to the acyl site. These results indicate that as in Sirt5, α5 plays a key role in Sirt4 acyl recognition. Residue differences seem to contribute directly to their different acyl selectivity profiles, but also to a more dynamic Sirt4 acyl site, which we speculate to adapt to acyl substrates (α5 N-terminus) and to contribute to a regulatory Sirt4-loop function (α5 center).

**Sirt4 substrates and structure enable modulator studies.** Our insights in Sirt4 catalytic activity and structure now enable activity studies for Sirt4 modulator development and a structure-based analysis of compound effects. Using the HMG-CPS1 peptide, we analyzed the effects of the Sirt1 activators resveratrol and SRT1720[39,40], the moderately Sirt1 selective inhibitors Ex527 and sirtinol[25,41], and the pan sirtuin inhibitor suramin[42,43]. At 10 and 100 μM compound concentration, respectively, there was no significant effect for resveratrol, Ex527, and sirtinol (Fig. 4a). Suramin caused potent Sirt4 inhibition, similar to its effects on other sirtuin isoforms[42,43], and 100 μM SRT1720 led to a weak signal decrease but with a high error and significant effect already in a control reaction, indicating a compound incompatibility with the coupled enzymatic assay used here. Suramin is a huge polyanionic compound occupying the complete active site of sirtuins and other targets[44]. Ex-527, in contrast, is accommodated by a small region around the C-site[25], and an overlay of Sirt3/Ex-527 with our Sirt4 structure rationalizes Sirt4's insensitivity (Fig. 4b): The compound would clash with its A ring with Pro200 from the Sirt4-loop, and omitting the A ring and instead attaching the carbamide via a methylene group to ring B might yield a Sirt4-specific compound.

As a convenient and complementary, fluorescence-based alternative to the coupled enzymatic assay with HMG-peptides, we established substrate and assay corresponding to the popular "Fluor-de-Lys" (FdL) deacylation assays[19]. Attaching the HMG group to a Lys with Z-protected amino group and 7-aminomethylcoumarin (AMC) coupled to the carboxyl group,

analog to the Sirt1–3 acetyl substrate Z-MAL[45], yielded an HMG-FdL substrate readily accepted by Sirt4 and sensitively monitored via fluorescence using the FdL procedure[19]. Testing suramin with the HMG-FdL substrate confirmed the potent Sirt4 inhibition observed in the coupled enzymatic assay (Fig. 4c), and a dose-response experiment yielded an $IC_{50}$ of $1.8 \pm 0.2\,\mu M$ (Supplementary Fig. 4a). Assays with SRT1720 showed no incompatibilities in control reactions and revealed that the compound has no effect on Sirt4 activity (Fig. 4c). We then employed the FdL-like Sirt4 assay for studying the effects of NAM and NADH, which are not compatible with the coupled assay. NAM appears to act as a physiological regulator for most sirtuin isoforms[37], and a NAM dose-response experiment with Sirt4 revealed potent inhibition with $IC_{50} = 13 \pm 2\,\mu M$ (Fig. 4d). The effect on Sirt4 is even more potent than on other isoforms[37,46] and possibly supported by the Sirt4-loop at the NAM accommodating C-site entrance. An NADH dose-response experiment (Fig. 4e), corrected through NADH spiking controls for its fluorescence overlap with AMC[47], also indicated pronounced Sirt4 inhibition ($IC_{50} = 126 \pm 12\,\mu M$ at 500 μM NAD$^+$). Analyzing the NADH titration with the robust MS assay confirmed this Sirt4 inhibition potency ($IC_{50} = 142 \pm 54\,\mu M$; Supplementary Fig. 4b) that exceeds NADH effects on other sirtuins ($IC_{50}$ 1.3–27.9 mM)[47,48], consistent with Sirt4 nucleotide site features (see above and discussion). The response of Sirt4 activity to NADH levels around ~30 μM, which is estimated to be the mitochondrial concentration of free NADH[47], suggests NADH or the NAD$^+$/NADH ratio to act as a physiological Sirt4 regulator.

## Discussion

Posttranslational modifications are a ubiquitous mechanism of protein regulation and rely on activated metabolites, which are now realized also to cause non-enzymatic modifications[14,28,29,49]. For emerging acyl modifications, such as succinylation and crotonylation, acyl-CoAs act as major activated metabolites[14] whose concentrations thus influence modification levels, together with deacylating enzymes. Elevated HMG-CoA levels in a HMG-CoA-Lyase deficiency model indeed increased protein HMG-ylation[28], and similar changes are expected under fasting conditions and during ketogenic protein catabolism, suggesting that de-HMG-ylating enzymes will regulate target functions during starvation. Sirtuins convert several acyl substrates but with isoform-specific selectivity profiles[15,38], and we find that Sirt4 also has a particular acyl preference profile. It shares de-HMG-ylation activity with Sirt5 but they differ, e.g., in their desuccinylase activity, and Sirt3–5 might have developed complementary selectivities to cover a range of acyls reflecting the variety of activated metabolites in mitochondria. Sirt5's higher de-HMG-ylation activity might suggest that other, even better Sirt4 substrate acyls might exist, but also shared deacylation activities can be complementary due to sirtuin differences in substrate sequence preference and tissue distribution[1,50]. The substrate acylations might be accidental "damage", which would render sirtuins repair enzymes, but many of them regulate dedicated target functions, consistent with specific regulatory effects of sirtuins[4,6]. The additional HMG-dependent NAD$^+$ hydrolase activity of Sirt4 could serve a signaling function, similar to NAD$^+$ depletion caused by poly-(ADP-ribose) polymerases, but a functional role of this unusual sirtuin activity remains to be studied further.

Sirtuin isoforms differ in acyl channel architecture and dynamics. Sirt5 provides a rather rigid binding site for succinylations, but related dicarboxylate modifications can adapt to this site, rationalizing Sirt5's activity against glutarylations[15,51] and HMG-ylations (present study). Sirt6, in contrast, has a long, hydrophobic channel to accommodate myristoyl substrates, and

also the smaller channels in Sirt2 and, in particular, Sirt3 can rearrange to efficiently accommodate and hydrolyze longer acylations such as myristoylations[23,38,52,53]. Sirt4 seems even more adaptable. Sirt4 structure and mutagenesis data suggest that, similar to Sirt5[13], the α5 center contributes to acyl recognition, but in Sirt4 even the remote α5 N-terminus influences acyl binding. Furthermore, the α5 center is connected via an Arg107/Asp203 salt bridge to the Sirt4-loop, an isoform-specific element of the Sirt4 acyl site, and the Sirt4-loop further to the nucleotide binding loop through the packing of Pro200 on the auto-inhibitory Tyr73. The Sirt4-loop assumes at least two states in our structures and for productive NAD[+] binding it indeed has to rearrange (Supplementary Figs. 2e, 3c)—possibly triggered by a yet to be identified activator in a physiological setting—which will induce conformational changes in the acyl site. Shifting Asp201 away from the conventional acyl binding pocket should support binding of dicarboxyl substrates such as HMG-ylations. A complete loop release, possibly, would open an area accommodating longer acyls in other isoforms[53] and could extend the observed Sirt4 deoctanoylation activity to longer modifications. Interestingly, Sirt4 accommodates extended lipoyl modifications as substrates despite its length restriction for fatty acids, likely by exploiting a Sirt4-specific channel branching off from the conventional acyl pocket. The bulkier lipoyl group might exploit this channel better than slim alkyl chains, and the two acyl sites might serve, alternatively or even in combination, to expand the acyl substrate spectrum of Sirt4. Interestingly, a lipoyl moiety would also fit into the Sirt4-specific channel coming from outside, as a target label in analogy to the bacterial sirtuin SirTM, which ADP-ribosylates targets only after their lipoylation[35], or as a potential regulatory metabolite (this study). Sirt4 substrate selection indeed appears dominated by $K_M$, and its de-HMG-ylation activity is still slightly lower than major activities of other sirtuin isoforms. Our comparison of HMG-peptides indicates a relevance for the substrate sequence, consistent with Sirt4 deacetylation studies[1], indicating that better de-HMG-ylation substrates might exist. Sequence differences might in fact account for our higher Sirt4-dependent delipoylation and Sirt5-dependent de-HMG-ylation activity compared to other studies[10,32]. It will be interesting to see whether better Sirt4 substrate sequences or acylations can be identified, and further mechanistic studies, supported by our structural data, should enable a full understanding of Sirt4-loop and acyl channel, their substrate adaptations, and possible external triggers that would regulate Sirt4.

Our Sirt4 substrates and assays enabled analyzes on drug effects and physiological Sirt4 regulators. The continuous coupled assay allows excellent quantification[27], and the FdL assay provides a sensitive set-up that is easily parallelized for screening campaigns[19]. Using these assays, we find a very potent NAM inhibition for Sirt4, which renders it the isoform most sensitive to NAM regulation so far[46,54]. Physiological NAM concentrations are assumed to reach up to 100 μM[54], which would inhibit Sirt4 almost completely, and in vivo Sirt4 activity will thus strongly depend on NAM levels. It will be interesting to see how Sirt4 contributes to the physiological effects of NAM. We further find that Sirt4 activity seems sensitive to physiological NADH levels or the NAD[+]/NADH ratio. Sirtuin regulation by NADH or NAD[+]/NADH had been suggested based on inhibitory NADH effects on other isoforms but was discarded due to its weak potency ($IC_{50} \geq 1.3$ mM; $K_i \geq 0.7$ mM), which rules out significant in vivo effects at assumed NADH concentrations (~30 μM in mitochondria) and NAD[+]/NADH ratios (10:1 and higher)[47,48]. We now find an at least one order of magnitude higher NADH sensitivity for Sirt4, which causes significant effects under such conditions, and it will be exciting to see how this Sirt4-specific regulation contributes to its function. Mechanistically, NADH

likely inhibits through nucleotide site binding in an extended, non-productive conformation also observed for NAD[+] under certain conditions[25] (Supplementary Fig. 3c). It places the NAM moiety outside the C-pocket—which prefers oxidized nucleotide due to its conserved negative charge—into regions that show positive electrostatic potential, in particular in Sirt4, which favors NADH over NAD[+]. Assays and structural insights also provide a basis for Sirt4 modulator development. The identified nucleotide binding site differences should enable Sirt4-specific inhibition, for example through Ex-527 derivatives that can enter Sirt4's tightened C-site. The particular Sirt4 acyl site could be exploited with thio-DMS-Lys, analog to the alkylimidate forming thio-acetyl peptides for deacetylases[25], and the additional Sirt4 channel provides another docking site for specific small molecule inhibitors and possibly also activators. Such Sirt4 modulators would be excellent tools for physiological studies and lead compounds for drug development, for example for diabetes treatment[7].

## Methods

**Chemicals.** If not stated otherwise, chemicals were from Sigma (St. Louis, MO, USA).

**Expression and purification of Sirt4 and Sirt5 proteins.** hSirt4(25–314) in pQE30 coding for an N-terminal His$_6$-tag, and xSirt4(32–315) and zSirt4(29–310) in a modified pET-19b coding for an N-terminal His$_6$-SUMO-tag (see Supplementary Table 1 for primer sequences), were expressed in *Escherichia coli* CodonPlus(DE3) (Agilent, Santa Clara, CA, USA) in LB medium supplemented with 100 μM zinc acetate. Cells were resuspended in 50 mM Tris/HCl pH 7.5, 200 mM NaCl (and 20% glycerol for xSirt4) and lysed by adding lysozyme and sodium desoxycholate and subsequent pressure homogenization in an Emulsiflex-C5 (Avestin, Ottawa, Canada). After centrifugation (75,000×*g*, 4 °C, 1 h), supernatants were incubated for 1 h with NiNTA beads in presence of 10 mM imidazole. After transfer in a column, the resin was washed with 20 column volumes (CV) 50 mM Tris/HCl pH 7.5, 500 mM NaCl and 20 CV 50 mM Tris/HCl pH 7.5, 200 mM NaCl, 10 mM imidazole, and the protein eluted with 50 mM Tris/HCl pH 7.5, 200 mM NaCl, 250 mM imidazole.

For hSirt4(25–314), the buffer of the eluted protein was changed to 50 mM Tris/HCl pH 7.5, 30 mM NaCl in a HiPrep Desalting Column (GE Healthcare, Chicago, IL, USA) and the sample applied to a SOURCE15S cation exchange column (GE Healthcare). The proteins were eluted in a linear gradient 30–1000 mM NaCl and Sirt4 fractions were pooled and diluted fivefold with 50 mM Tris/HCl pH 7.5. Sirt4 was then loaded on a HiTrap Heparin column (GE Healthcare) and eluted with a linear gradient 30–1000 mM NaCl. The protein was concentrated to ~1 mg ml[−1] in a Microsep concentrator (Pall Corporation, Port Washington, NY, USA), flash-frozen in liquid nitrogen, and stored at −80 °C.

For xSirt4(32–315) and zSirt4(29–310) eluted from the NiNTA material, the buffer was changed to 50 mM Tris/HCl pH 7.5, 200 mM NaCl, (supplemented with 20% glycerol for xSirt4) in a HiPrep Desalting column and subjected to Senp2-proteolysis for 30 min on ice. The protein was loaded on a HisTrap HP 1 mL column (GE Healthcare) and eluted with 50 mM Tris/HCl pH 7.5, 200 mM NaCl, 30–50 mM imidazole. It was subsequently subjected to gel filtration on a Superdex 75 10/300 GL (xSirt4) or Superdex 200 10/300 GL (zSirt4) column (both GE Healthcare) in 25 mM Tris/HCl pH 7.5, 150 mM NaCl (+20 % glycerol for xSirt4), concentrated in a Microsep concentrator, flash-frozen in liquid nitrogen, and stored at −80 °C. xSirt4 single-site variants were generated using the QuickChange protocol (see Supplementary Table 1 for primer sequences) and verified by DNA sequencing, and the proteins were produced as described for wild-type xSirt4.

zSirt5(30–298) was expressed in *E.coli* and purified through Talon affinity chromatography, TEV-proteolysis, reverse affinity chromatography, and gel filtration on a Sephacryl S-200 column[15].

**Synthesis of Lys- and peptide-based substrates.** HMG-ylated peptides CPS1 (Bz-GVL(acyl-K)EYGV-NH2), MCD (Ac-TSYLGS(HMG-K)IIKASE-NH2), and NNT (Ac-NITKLL(HMG-K)AISPDK-NH2) were synthesized using Fmoc-based solid-phase peptide synthesis protocols. Fmoc-Lys(Nosyl)-OH was used as building block enabling selective on resin deprotections and acylations using HMG-anhydride. As acyl peptide library, the panel of acyl CPS1-Lys527 peptides described in ref. [15] was used and extended through analog synthesis of β-Ala-CPS1, γ-aminobutyryl-CPS1, 6-aminocaproyl-CPS1, butyryl-CPS1, octanoyl-CPS1, lipoyl-CPS1, and biotinyl-CPS1.

The Sirt4 substrate Z-Lys(HMG)-AMC (MC3659; 5-(((S)−5-(((benzyloxy)carbonyl)amino)-6-((4-methyl-2-oxo-2H-chromen-7-yl)amino)-6-oxohexyl)amino)-3-hydroxy-3-methyl-5-oxopentanoic acid) was synthesized by reaction between the (S)-benzyl (6-amino-1-((4-methyl-2-oxo-2H-chromen-7-yl)amino)-1-oxohexan-2-yl)carbamate (Z-Lys-AMC), prepared as reported in literature[55], and

the commercially available 3-hydroxy-3-methylglutaric (HMG) anhydride in dry THF in the presence of DIPEA at room temperature (Supplementary Fig. 5). All chemicals were purchased from Aldrich Chimica, Milan (Italy), and were of the highest purity. Z-Lys-AMC (170 mg, 0.388 mmol) was dissolved at 0 °C in 4 mL dry THF together with diisopropylethylamine (DIPEA) (140 μL, 0.777 mmol) under nitrogen atmosphere. A solution of 3-hydroxy-3-methylglutaric anhydride (67 mg, 0.777 mmol) in dry THF (4 mL) was added dropwise at 0 °C, and the resulting mixture was left under stirring at room temperature overnight. At the end of the reaction, water (10 mL) was added, the resulting mixture was made acidic (pH ∼2) with potassium bisulphate 1 M and then extracted with ethyl acetate (6 × 10 mL). The collected organic phases were washed with brine (2 mL), dried, and concentrated under reduced pressure to provide a crude residue that was purified by a silica gel flash chromatography (SNAP 25, Biotage Isolera One) using a linear gradient of methanol (3–25%) in chloroform, giving the expected compound Z-Lys (HMG)-AMC as a white solid, with a yield of 72%. Melting point was determined on a Buchi 530 melting point apparatus and is uncorrected. $^1$H- and 13C-NMR spectra were recorded at 400 MHz ($^1$H) respective 100 MHz ($^{13}$C) on a Bruker AC 400 spectrometer; reporting chemical shifts in δ (p.p.m.) units relative to the internal reference tetramethylsilane (Me$_4$Si). All compounds were routinely checked by TLC and $^1$H-NMR. TLC was performed on aluminum-backed silica gel plates (Merck DC, Alufolien Kieselgel 60 F254) with spots visualized by UV light. Yield of reaction refers to the purified product. Mass spectrum was recorded on an API-TOF Mariner by Perspective Biosystem (Stratford, TX, USA), and samples were injected by a Harvard pump, using a flow rate of 5–10 μL min$^{-1}$, in the Electrospray system. Elemental analysis was performed by a PE 2400 (Perkin-Elmer) analyzer and has been used to determine purity of the described compound, which is >95%. Analytical result is within ±0.40% of the theoretical value.

mp 128–129 °C; yield 72%. $^1$H-NMR (DMSO) δ 1.18 (s, 3H, C(OH)CH$_3$), 1.32–1.41 (m, 4H, CHCH$_2$CH$_2$CH$_2$NH), 1.61–1.67 (m, 2H, CHCH$_2$CH$_2$CH$_2$NH), 2.33 (s, 2H, C(OH)(CH$_3$)CH$_2$CONHCH$_2$), 2.38–2.41 (m, 5H, CH$_2$COOH and coumarin-CH$_3$), 3.04 (m, 2H, CH$_2$CONHCH$_2$), 4.15 (m, 1H, α-CH), 5.04 (s, 2H, OCH$_2$Ph), 6.28 (s, 1H, C–H coumarin), 7.18–7.37 (m, 5H, C–H phenyl ring), 7.51–7.54 (m, 1H, C–H coumarin), 7.67 (m, 1H, C–H coumarin), 7.73 (d, 1H, Z–NHCHCO), 7.80 (bs, 1H, C–H coumarin), 8.01 (t, 1H, CH$_2$CONHCH$_2$), 10.61 (bs, 1H, CONH-coumarin). $^{13}$C-NMR (DMSO) δ: 174.2, 172.6, 170.9, 161.4, 156.5, 154.2, 153.5, 142.5, 136.8, 128.6 (2C), 128.2 (2C), 127.8, 124.5, 116.7, 112.5, 110.7, 105.2, 71.5, 66.8, 54.1, 47.6, 46.9, 39.1, 30.5, 29.7, 27.4, 22.5, 18.9. Anal. (C$_{30}$H$_{35}$N$_3$O$_9$) Calcd. (%): C, 61.95; H, 6.07; N, 7.22. Found (%): C, 62.03; H, 6.05; N, 7.19. MS (ESI), m/z: 580 (M–H)$^-$.

**CypA modification and de-HMG-ylation and intact protein MS.** The recombinant CypA was prepared through expression of a full-length CypA construct in pQE70 (Qiagen) in *E. coli* M15 cells and purification by ion exchange chromatography on Fractogel EMD DEAE-650(M), Fractogel TSK AF-Blue, and Fractogel SO3–650(M) (Merck Millipore)[29]. CypA protein (0.4 mg ml$^{-1}$) was modified using 8 mM HMG-CoA in 100 mM Tris-HCl buffer pH 8.3 at 37 °C for 4 h. Formation of HMG-ylated protein was confirmed by peptide-MS analyzes after tryptic digest[29], and by intact HMG-CypA mass analyzes through HPLC-coupled ESI-MS[56]. For intact mass determination, 25 μM HMG-CypA in 10 mM Tris/HCl pH 7.5, 50 mM NaCl was HPLC-separated with the setting described below for peptide MS, using a 30 min gradient from 1 to 55% buffer B (90% ACN, 9.9% H$_2$O, 0.1% FA; buffer A: 5% ACN, 94.9% H$_2$O, 0.1% FA) followed by 1 min of 55–90% buffer B with 70 μL min$^{-1}$ flow-rate. MS-analyzes were done with the settings described for peptide assays (below), except that the IntactProtein script was activated, which reduces CEM to 100. Acquired data were extracted with PeakView and deconvoluted in MassLynx in the range of 950–1500 m/z using the MaxEnt I algorithm to a resolution of 1 Da. Peak intensity values were recorded and overlaid with MassLynx.

For MS-based CypA deacylation analyzes, 50 μM HMG-modified CypA was incubated for 2 h at 37 °C with 20 μM hSirt4 and 2 mM NAD$^+$, as well as 0.5 mg mL$^{-1}$ nicotinamidase to prevent Sirt4-inhibition by released nicotinamide. Control reactions without Sirt4 were incubated for 22 h. All reactions were stopped through mixing 1:1 with 0.5% TFA and analyzed by ESI-MS as described above[56]. CypA de-HMG-ylations in the coupled continuous assay were performed as described for peptide-based assays.

**Peptide- and FdL-based activity assays.** The coupled continuous assay was performed as reported[27]. Briefly, assays in 20 mM sodium phosphate buffer pH 7.8 contained 5 μM hSirt4 or 3–5 μM xSirt4, 0.05 mg ml$^{-1}$ nicotinamidase, 2 U ml$^{-1}$ GDH, 3.3 mM a-ketoglutarate, 0.2 mM NADPH, 10% DMSO, and NAD$^+$ at 2 mM or as indicated, and substrate peptide at 500 μM or as indicated. Reactions were monitored in microplates at room temperature for 1 h through absorption measurements at 340 nm in a LAMBDAScan plate reader (MWG Biotech, Ebersberg, Germany).

The Fluor-de-Lys (FdL) assay was performed at 37 °C in 25 mM Tris/HCl, 150 mM NaCl with 1 μM hSirt4, 500 μM HMG-FdL substrate and 500 μM NAD$^+$. After 20 min, developer solution (2 mM NAM and 10 mg mL$^{-1}$ trypsin) in assay buffer was added 1:1 and samples were incubated for 45 min at room temperature. Fluorescence was measures using a FluoDia T70 with excitation wavelength 365 nm and emission wavelength 465 nm.

For MS and UV analyzes of deacylation reactions, samples were prepared as described for the coupled continuous assay, stopped by mixing 1:1 with 0.5% TFA after 0 and 60 min, diluted to 20 μM peptide using 0.1% FA, and analyzed by HPLC-separation using a Shimadzu Prominence UFLC (Shimadzu, Duisburg, Germany) coupled to ESI-MS and UV detection. Samples were washed on a Piccolo Proto 200 C4 5 μm 2.5 × 0.5 mm trap column (Higgins Analytical, Mountain View, California) and subsequently subjected to a Jupiter C4 5 μm 300 Å 150 × 1 mm analytical column (Phenomenex, Torrance, California) for reversed phase separation, with 99.9% H$_2$O, 0.1% TFA as buffer A and 99.9% ACN, 0.1% TFA as buffer B. Peptides were eluted over a 20 min gradient from 1 to 55% buffer B, followed by 1 min from 55 to 90% buffer B with 70 μL min$^{-1}$ flow-rate. UV-detection was done using a Shimadzu SPD-20A detector at 280 nm. MS analysis was performed by ESI-TOF-MS on an AB Sciex TripleTOF 5600+ mass spectrometer (Sciex, Darmstadt, Germany) with a DuoSpray Ion Source using the following settings: floating voltage of 5500 V and declustering potential of 100 with one TOF experiment summing over four time bins. We detected in a mass range from 300 to 2500 m/z. XIC of substrate and product peptides were extracted using their respective mass in singly, doubly, or triply charged state within a window of 0.2 m/z in PeakView version 1.2.0.3.

**Thermal denaturation shift assays.** Thermal shift assays were performed in 96-well PCR plates using 3 μM xSirt4 (32–315) or 3 μM zSirt4 (29–310) and SYPRO orange (Thermo Fisher, Waltham, MA, USA) covered with 15 μL mineral oil. Heating and fluorescence measurements were performed in a FluoDia T70 with 1 K steps from 20 to 73 °C (excitation: 465 nm, emission: 580 nm). The data were analyzed in GraFit (Erithacus Software Ltd, Horley, UK) by nonlinear fitting using a two state model.

**Statistical information.** Data points for activity assays were determined in duplicates, and all experiments were done in at least two repetitions.

**Sirt4 crystallization and structure determination.** xSirt4(32–315) protein (5 mg ml$^{-1}$ in 25 mM Tris/HCl 7.5, 150 mM NaCl, 20% glycerol) was incubated with 10 mM ADPr (xSirt4/ADPr complex) or 1 mM thioacetyl H3K9 peptide and 5 mM NAD$^+$ (xSirt4/thioacetyl-ADPr complex) on ice for 30 min. The xSirt4 complexes crystallized at 4 °C in sitting drops (1:1 ratio protein to reservoir solution) with 100 mM BICINE pH 8.5, 20% PEG6000 (xSirt4/ADPr) or 500 mM Na/K-tartrate, 0.5% PEG5000MME, 100 mM TRIS/HCl pH 8.5 (xSirt4/thioacetyl-ADPr) as reservoir. zSirt5 (10 mg mL$^{-1}$ in 20 mM TRIS/HCl pH 8.5, 150 mM NaCl) was incubated with 1 mM HMG-CPS1-peptide (10% v/v DMSO final concentration) on ice for 30 min and crystallized in sitting drops at 20 °C using 20% PEG3350, 100 mM HEPES pH 7.6 as reservoir solution. Crystals appeared within 3 days (xSirt4) or 2 weeks (zSirt5), were transferred to a drop of reservoir solution supplemented with the co-crystallization ligand and 25% glycerol for 1 min, and flash-frozen in liquid nitrogen.

Data collection was done at BESSY II beamline MX14.1 (operated by the Helmholtz Zentrum Berlin, Germany) using a wavelength of 0.912 Å and a Pilatus 6 M detector (Dectris, Baden, Switzerland). Indexing, scaling and merging of diffraction data was performed with XDS[57]. Structures were solved by molecular replacement with PHASER[58] using as a search model a Sir2Af1/peptide complex (PDB-Code 4TWI[59]) for xSirt4/ADPr, this initial xSirt4 structure for xSirt4/thioacetyl-ADPr, and a zSirt5/peptide complex (PDB ID 4UTV[15]) for zSirt5/HMG-CPS1. Refinement was performed with Refmac5[60], using partially anisotropic B-factors and TLS for the protein chain and the Zn ion for the xSirt4/ADPr complex. Models were built and validated using Coot[61] and structure figures were generated with PyMol (Schrödinger, LLC). Docking of Lipoyl-Lys in the xSirt4/ADPr active-site was done with LeadIT (BiosolveIT GmbH, Sankt Augustin, Germany).

**Sequence alignments and phylogenetic trees.** Structure-based sequence alignments were created using STRAP with the integrated Aligner3D algorithm[62], and manual editing and phylogenetic tree generation were done with BioEdit[63]. Conservation levels were mapped on the xSirt4 structure surface using ConSurf[64].

**Data availability.** Structure factors and refined structures have been deposited with the Protein Data Bank (http://www.rcsb.org/pdb) under accession codes 5OJ7 (xSirt4/ADPr), 5OJN (xSirt4/thioacetyl-ADPr), and 5OJO (zSirt5/HMG-CPS1). Other data are available from the corresponding author upon reasonable request.

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

## Acknowledgments

We thank Dr Eric Verdin for helpful discussions, Dr Cordelia Schiene-Fischer for providing cyclophilin A, and the beamline staff of BESSY for technical support. This work was

supported by grants from Oberfrankenstiftung (to CSt); EU COST-Action: EPICHEMBIO (CM1406), FP7 BLUEPRINT (contract n° 282510), A-PARADDISE (contract n° 602080), PRIN 2016 (Prot. 20152TE5PK), AIRC 2016 (Project n. 19162; to A.M.); PRIN 2012 (Prot. 2012CTAYSY), AIRC Fondazione Cariplo TRIDEO Id. 17515 (to D.R.).

## Author contributions

M.P. and C.S. designed the project, analyzed data, and drafted the manuscript, and all authors contributed to the refinement of the manuscript. M.P. solved the crystal structures and did the activity studies. M.S., Z.S., M.P. and M.F. did acylation studies, and M.S. and M.M. synthesized acyl peptides. D.R. and A.M. created the fluorogenic Z-Lys (HMG)-AMC substrate.

## Additional information

**Competing interests:** The authors declare no competing financial interests.

