## [Peer Review file · Nature Communications]

Reviewers' comments:

Reviewer #1 (Remarks to the Author):

The manuscript on Sirt4 by Steegborn and coworker contain several interesting findings: novel more efficient activity, a crystal structure (which I believe is the first one for Sirt4), nicotinamide and NADH as efficient inhibitors of Sirt4. Sirt4's activity has been a puzzle in the sirtuin community and thus this study has some important contributions. For this reason, I support the publication of this manuscript. The physiological significance of the novel modifications has recently been demonstrated in another publication recently from another lab, but I do not think the other publication undermines this work, which has more structural component. However, I do have several criticisms that I think the authors should try to address before publishing.

1. The most important scientific criticism is that the crystal structure does not provide a satisfactory explanation for the acyl group specificity of Sirt4. Part of the reason, of course, is that the authors could not obtain the crystal of Sirt4 with HMG peptide. Instead, they have a structure of Sirt5 with HMG peptide and they tried to rationalize the acyl group specificity of Sirt4 by superimpose the two structures. This analysis is fine except that I found it dissatisfying that none of the mutations, including Asp201Ala, Asp203Ala, caused much effect to the catalytic activity. Furthermore, even deleting the loop does not affect the activity much. Intellectually, this does not make much sense.

2. The writing needs significant improvement. In many places, I found the sentences are difficult to understand. For example: In the abstract "They further identify a conserved, isoform-specific Sirt4 loop that folds into the active site to regulate catalysis", I think "they" should be "we" and "regulating catalysis" is not accurate as it is not demonstrated in this manuscript. Page 12, "In the region accommodating a distal acyl group in short and medium long diacyls", "diacyls" is not an accurate term as it could mean two acyl groups attached to lysine. Page 17, "and the FdL assay provides a sensitive setup with complementary incompatibilities that is easily parallelized for screening campaigns", this sentence is hard to understand, especially "complementary incompatibilities". This is just a few examples. I think the manuscript would benefit a lot from a more careful scrutiny.

3. On page 16, the authors states that "Sirt6 has a long, hydrophobic channel to accommodate myristoyl substrates, but also the smaller channels in Sirt2 and, in particular, Sirt3 can rearrange to accommodate myristoylations." I think more papers should be cited here regarding the ability of Sirt1-3 to remove long chain fatty acyl groups. The labs of John Denu and Hening Lin have several papers demonstrating that Sirt1-3 has this ability.

Reviewer #2 (Remarks to the Author):

The manuscript by Steegborn and coworkers entitled: "Crystal structure of the mitochondrial deacylase Sirtuin 4 – unique structural features, acyl selectivity, and regulation", describes the first X-ray crystal structure of sirtuin 4 as well as biochemical evaluation in vitro.

Sirtuins are important regulators of epigenetic states and metabolism among others, and sirtuin 4 has been difficult to characterize in vitro thus far. Though, not a human enzyme, the authors do a convincing job of phylogenetic and sequence comparisons to argue the importance of the structural insight gained from the X-ray crystal structure of *Xenopus* sirtuin 4. This is an important study with high potential impact, which will help guide future drug discovery studies aimed at this enzyme.

The work should be of broad interest to the readership and I therefore recommend acceptance for publication in *Nat. Commun.* after revision as outlined below.

1) The substrate scope and related structural features should be discussed in the context of the recent publication on SIRT4 from the Hirschev laboratory, *Cell Metab.* 25, 838, 2017

2) Likewise, the discussion on non-enzymatic acylation should include reflections of recently

published works as well, such as *Cell Metab.* 25, 823, 2017 and *Cell Chem. Biol.* 24, 231, 2017.

3) In the discussion of classification, only the deacetylase activity of sirtuin 2 is mentioned, but it has been shown kinetically in several publications now that this enzyme is much more active towards longer chains. This should be included in the discussion.

4) Also, the authors nicely speculate on possible evolutionary developments related to HATs, but is it known which enzymes existed first?

5) The authors correctly state that their continuous assay is incompatible with NADH inhibition experiments, however, neither is the "FdL" assay used by the authors. NADH fluorescence overlaps with that of AMC, and the presented inhibition data are therefore not valid. Thus, the section on NADH inhibition should either be deleted or redone using another assay format.

Reviewer #3 (Remarks to the Author):

Summary:

Although the mitochondrial sirtuin, SIRT4, undoubtedly plays central roles regulating fuel utilization and metabolism in the mitochondria, knowledge of the substrate acyl preference of SIRT4 has lagged far behind the biology. In this manuscript, Pannek et al. screened the activity of several SIRT4 homologues against a library of peptides containing different acyl modifications, and identified HMG-ylation as a new deacylating activity for SIRT4. The authors then determined structures of SIRT4 from *X. tropicalis* bound to ADPribose and thioacetyl-ADPribose, revealing a loop specific to SIRT4 that bridges the Zn finger and Rossmann fold domains. As the structure of SIRT4 and its acyl specificity represent fundamental gaps in PTM-mediated regulation of mitochondrial metabolism, this manuscript has the potential to impact many different areas of mitochondrial biology. However, several major issues need to be addressed in order to demonstrate that SIRT4 robustly de-HMG-ylates peptides and that structural features observed in the SIRT4 crystal structure regulate catalysis by SIRT4.

Major Issues:

1. This manuscript relies on an enzyme-coupled assay to measure SIRT4 activity, which provides an indirect read-out of deacylation. Since the assay monitors NAD⁺ consumption (nicotinamide production), it is possible that these alternate acyl substrates stimulate NAD⁺ turnover, but not coupled to peptide deacylation. To confirm that the new deacylating activities attributed to SIRT4 actually represent new substrate specificities, the authors should confirm their steady-state kinetic measurements using an assay that monitors peptide deacylation directly (i.e. HPLC-based read-out) for at the very least acetyl-CPS1, lipoyl-CPS1, and HMG-CPS1.

2. The authors should clarify the conditions used for the screen in the text of the paper (was it done with the continuous coupled assay? What concentration of peptide/NAD⁺?)

3. Likewise, SIRT4-dependent de-HMG-ylation of CypA should be confirmed directly by mass spectrometry.

4. In figure 2b, the SIRT4 loop looks like it packs against Tyr73 (via Pro200). Although the authors explore the possibility that the loop contributes to peptide binding, they do not check whether it affects binding of NAD⁺ to the enzyme. Given that this loop is a major structural difference between SIRT4 and other sirtuins, the authors should check whether deleting the loop alters apparent affinity for NAD⁺.

5. The second distinguishing feature of the SIRT4 active site is the channel that branches off from

the acyl-lysine binding tunnel and reaches the surface of the enzyme. This raises the possibility that a small molecule may stimulate the activity of SIRT4 (similar to SIRT6, where free fatty acids stimulate deacetylation activity JBC 2013). The authors should test whether adding free fatty acids, lipoic acid, or ketone bodies (i.e. beta-hydroxybutyrate or acetoacetate) stimulates SIRT4 activity against acetylated peptides, as well as HMG-ylated peptides.

6. It is hard to reconcile the fact that HMG-ylation is removed far faster by SIRT5 than SIRT4 under the conditions tested in Fig. 1D, with the apparent specificity of SIRT4 for HMG-ylated peptides. If a full steady-state kinetic titration is done with SIRT5 and an HMG-ylated substrate, how does it compare to SIRT4-catalyzed de-HMG-ylation? In other words, does SIRT4 have higher k_{cat} for HMG-ylated peptides as compared to SIRT5? What is the overlap between SIRT4 and SIRT5 deacylating activities?

7. Accession codes need to be added from the PDB.

Minor Issues:

1. In Fig. 1C, it would be helpful to expand the low-substrate portion of the graph to show the differences in K_M between the different peptides.

2. The authors should report $CC_{1/2}$ and CC^* statistics in Table 2.

3. Do crystal contacts influence the conformation of the SIRT4-loop in the complex with ADPribose?

4. In figure 3E/3F it would be helpful if the HMG-ylated peptide is displayed in a different color from the rest of the protein.

5. It would be interesting if the authors could speculate in the discussion about the biological significance of de-HMG-ylation by SIRT4 (i.e. conditions where ketone body synthesis increases such as fasting, connections to cholesterol synthesis via the mevalonate pathway, etc. and regulation by SIRT4).

Changes made in response to referee comments

Reviewer #1 (Remarks to the Author):

The manuscript on Sirt4 by Steegborn and coworker contain several interesting findings: novel more efficient activity, a crystal structure (which I believe is the first one for Sirt4), nicotinamide and NADH as efficient inhibitors of Sirt4. Sirt4's activity has been a puzzle in the sirtuin community and thus this study has some important contributions. For this reason, I support the publication of this manuscript. The physiological significance of the novel modifications has recently been demonstrated in another publication recently from another lab, but I do not think the other publication undermines this work, which has more structural component. However, I do have several criticisms that I think the authors should try to address before publishing.

1. The most important scientific criticism is that the crystal structure does not provide a satisfactory explanation for the acyl group specificity of Sirt4. Part of the reason, of course, is that the authors could not obtain the crystal of Sirt4 with HMG peptide. Instead, they have a structure of Sirt5 with HMG peptide and they tried to rationalize the acyl group specificity of Sirt4 by superimpose the two structures. This analysis is fine except that I found it dissatisfying that none of the mutations, including Asp201Ala, Asp203Ala, caused much effect to the catalytic activity. Furthermore, even deleting the loop does not affect the activity much. Intellectually, this does not make much sense.

We agree that our first Sirt4 structures provide only partial insight in Sirt4's apparently still incompletely revealed substrate specificity, similar to many other systems where several structures in combination with biochemical experiments over many years and publications were required for a complete understanding. We feel, however, that our structural and biochemical data already provide several important insights, on aspects such as NAD recognition, evolution and sequence preference but also with respect to substrate acyl selection, including the additional active site channel, the non-equivalence of the active site Y-XX-R motif to the Sirt5 acyl recognition motif, and the arrangement and apparent adaptability of alpha5. We slightly changed results and discussion to make these points more visible (e.g., page 7,11,18). Furthermore, the structures will be very important for the design and interpretation of future studies on this topic (now clearly stated on page 19).

We understand the referee's disappointment about the limited effects of several mutations, including conserved loop residues and the deletion lacking the entire Sirt4 loop. However, some of our mutations did have significant effects, and we revised the text slightly to make this fact more visible (page 13, 15). Also, the lack of more dramatic effects despite an observed loop position that tightens active site pockets indicates that additional factors that influence this conformation to modulate Sirt4 activity are missing. We modified the text to make this point clearer (page 18).

2. *The writing needs significant improvement. In many places, I found the sentences are difficult to understand. For example: In the abstract "They further identify a conserved, isoform-specific Sirt4 loop that folds into the active site to regulate catalysis", I think "they" should be "we" and "regulating catalysis" is not accurate as it is not demonstrated in this manuscript. Page 12, "In the region accommodating a distal acyl group in short and medium long diacyls", "diacyls" is not an accurate term as it could mean two acyl groups attached to lysine. Page 17, "and the FdL assay provides a sensitive setup with complementary incompatibilities that is easily parallelized for screening campaigns", this sentence is hard to understand, especially "complementary incompatibilities". This is just a few examples. I think the manuscript would benefit a lot from a more careful scrutiny.*

We followed the suggestion of the referee and carefully checked the language of the manuscript:

Abstract: "They further identify ..." was changed to "The structures further identify ..."

Abstract, Introduction: We added "potentially" (Abstract and page 4) to clarify that a regulatory role for the Sirt4-loop is suggested but not fully confirmed and characterized by the available data.

We corrected "diacyl" to "dicarboxylate" on page 13.

We simplified the sentence in the discussion of assays (page 19) by removing the term "with complementary incompatibilities".

In addition, we slightly changed the wording in the following sections to make them easier to read:

Comparison of the acyl selectivities of Sirt3,4,5 (page 6-7; see also Referee #3, Top #6).

Effects of mutations (page 15; see above, Top #1)

Longer chain deacylation activity of Sirt2 and Sirt3 (page 18; see also Referee #2, Top #3)

Discussion on substrate sequence relevance (page 19)

3. *On page 16, the authors states that "Sirt6 has a long, hydrophobic channel to accommodate myristoyl substrates, but also the smaller channels in Sirt2 and, in particular, Sirt3 can rearrange to accommodate myristoylations." I think more papers should be cited here regarding the ability of Sirt1-3 to remove long chain fatty acyl groups. The labs of John Denu and Hening Lin have several papers demonstrating that Sirt1-3 has this ability.*

We now added a reference from the Lin lab on the structural and enzymatic characterization of the acyl channel in Sirt2 (Teng et al., Sci Rep 2015) to the previous citations, which already included a reference to the detailed enzymatic Sirt2/3 work and structural Sirt2 analysis from the Denu lab (Feldman et al., Biochemistry 2015). The already included reference Gai et al. (FEBS Lett 2016) comprises similar structural analyses on Sirt3. While additional, less detailed enzymatic studies on acyl substrates from several groups preceded these papers, we have to restrict the number of references to meet journal restrictions but also to guide the reader to the most relevant work for our claims. We feel that the chosen references (which include appropriate citations to the previous studies) are the key mechanistic studies on this aspect and most appropriate in context of the statement we make and now offer a fair and helpful reference to previous work.

Reviewer #2 (Remarks to the Author):

The manuscript by Steegborn and coworkers entitled: "Crystal structure of the mitochondrial deacylase Sirtuin 4 – unique structural features, acyl selectivity, and regulation", describes the first X-ray crystal structure of sirtuin 4 as well as biochemical evaluation in vitro. Sirtuins are important regulators of epigenetic states and metabolism among others, and sirtuin 4 has

been difficult to characterize in vitro thus far. Though, not a human enzyme, the authors do a convincing job of phylogenetic and sequence comparisons to argue the importance of the structural insight gained from the X-ray crystal structure of Xenopus sirtuin 4. This is an important study with high potential impact, which will help guide future drug discovery studies aimed at this enzyme. The work should be of broad interest to the readership and I therefore recommend acceptance for publication in Nat. Commun. after revision as outlined below.

1) The substrate scope and related structural features should be discussed in the context of the recent publication on SIRT4 from the Hirschey laboratory, Cell Metab. 25, 838, 2017

The publication from the Hirschey lab indeed supports our conclusion that the HMG-modification is a Sirt4 substrate. We had already mentioned in our manuscript Hirschey's work confirming the physiological occurrence and relevance of this modification, which nicely complements our structural and mechanistic work, and we have now changed the citation from "in press" to the recently released publication (page 8). We further added a discussion of Hirschey's and our results concerning the ability of Sirt4 and Sirt5 to remove HMG modifications, in particular structural aspects and potential substrate sequence effects (page 18, 19; see also referee #3, top #6).

2) Likewise, the discussion on non-enzymatic acylation should include reflections of recently published works as well, such as Cell Metab. 25, 823, 2017 and Cell Chem. Biol. 24, 231, 2017.

We followed the suggestion to add references and short additions based on Wagner *et al.* (Cell Metab. 25, 823) to results and discussion (page 7, 8, 17).

Although being quite interesting, we refrained from adding the second suggested reference, since we feel that we cannot cover all aspects and publications on the booming topic of non-enzymatic protein acylation and consider this particular work, which is more focused on describing an elegant technique for studying non-enzymatic acylation, not essential for the points we make.

3) In the discussion of classification, only the deacetylase activity of sirtuin 2 is mentioned, but it has been shown kinetically in several publications now that this enzyme is much more active towards longer chains. This should be included in the discussion.

We slightly changed our discussion on the adaptability of the Sirt2 and Sirt3 acyl channels to clearly state that longer acyl modifications such as myristoylations can also be efficiently hydrolyzed by these isoforms (page 18), and we added another reference on these enzymatic and structural studies to our statement. See also Referee #1, Top #3.

4) Also, the authors nicely speculate on possible evolutionary developments related to HATs, but is it known which enzymes existed first?

We agree that this is a very interesting question related to the topic of the evolution of deacylases, but we are not aware of studies providing clear data suggesting a particular order of emergence of HATs and HDACs. Based on our discussion in the text that non-enzymatic modifications appear to have preceded the modifying enzymes, we would speculate that the need for removal of the modifications has emerged first. However, since we provide no clear evidence for this in our manuscript and are also not aware of publications we can refer to, we feel it would be better not to extend the discussion in the present manuscript to this question.

5)The authors correctly state that their continuous assay is incompatible with NADH inhibition experiments, however, neither is the "FdL" assay used by the authors. NADH fluorescence overlaps with that of AMC, and the presented inhibition data are therefore not valid. Thus, the section on NADH inhibition should either be deleted or redone using another assay format.

The referee is of course right that NADH fluorescence overlaps with AMC fluorescence, which is well documented in the work by Madsen et al. (J. Biol. Chem. 2016) cited in our manuscript. However, careful controls with spikes of the same NADH concentration after the developer step, in regular assays as well as control reactions, confirmed that the NADH signal is additive and can be corrected for. Nevertheless, we fully agree with the referee that it would be appropriate to confirm our finding in a Sirt4 assay that is insensitive to any interference from NADH. We therefore repeated the inhibition experiments with a mass spectrometry assay, which indeed confirmed our previous result. We added a comment on the fluorescence overlap between NADH and FdL, with a reference to Madsen et al. (J. Biol. Chem. 2016), and the new mass spectrometry-based inhibition data (page 16-17; Supplementary Fig. 4b).

Reviewer #3 (Remarks to the Author):

Summary:

*Although the mitochondrial sirtuin, SIRT4, undoubtedly plays central roles regulating fuel utilization and metabolism in the mitochondria, knowledge of the substrate acyl preference of SIRT4 has lagged far behind the biology. In this manuscript, Pannek et al. screened the activity of several SIRT4 homologues against a library of peptides containing different acyl modifications, and identified HMG-ylation as a new deacylating activity for SIRT4. The authors then determined structures of SIRT4 from *X. tropicalis* bound to ADPribose and thioacetyl-ADPribose, revealing a loop specific to SIRT4 that bridges the Zn finger and Rossmann fold domains. As the structure of SIRT4 and its acyl specificity represent fundamental gaps in PTM-mediated regulation of mitochondrial metabolism, this manuscript has the potential to impact many different areas of mitochondrial biology. However, several major issues need to be addressed in order to demonstrate that SIRT4 robustly de-HMG-ylates peptides and that structural features observed in the SIRT4 crystal structure regulate catalysis by SIRT4.*

Major Issues:

1. This manuscript relies on an enzyme-coupled assay to measure SIRT4 activity, which provides an indirect read-out of deacylation. Since the assay monitors NAD⁺ consumption (nicotinamide production), it is possible that these alternate acyl substrates stimulate NAD⁺ turnover, but not coupled to peptide deacylation. To confirm that the new deacylating activities attributed to SIRT4 actually represent new substrate specificities, the authors should confirm their steady-state kinetic measurements using an assay that monitors peptide deacylation directly (i.e HPLC-based read-out) for at the very least acetyl-CPS1, lipoyl-CPS1, and HMG-CPS1.

Although the Sirt4 de-HMG-ylation activity is also documented in our manuscript by the FdL-like assays, whose signal is generated via the deacylated product peptide, we have to thank the referee for his insightful and constructive criticism on this topic (and others below). It prevented us from overlooking a new aspect of Sirt4 enzymology, HMG-stimulated NAD⁺ hydrolysis, which has been missed by the parallel work from the Hirschey lab. We included the key measurements, done with a mass spectrometry assay detecting substrate and product peptide, which show that Sirt4 features deacylation as well as stimulated NAD⁺ hydrolase activity with HMG-modified peptide, while it act solely as a deacylase with acetyl substrate (page 6, 18). See also top #3.

2. The authors should clarify the conditions used for the screen in the text of the paper (was it done with the continuous coupled assay? What concentration of peptide/NAD⁺?)

All information on assays and substrate concentrations is included in the Methods section, which was placed at the end of the initial manuscript based on the formatting guidelines of Nature Structural

Biology. We have now moved the Methods section behind the discussion, and the requested information on assay conditions is thus now part of the regular manuscript text.

3. Likewise, SIRT4-dependent de-HMG-ylation of CypA should be confirmed directly by mass spectrometry.

As suggested, we included a mass spectrometry experiment to confirm Sirt4-dependent de-HMG-ylation of CypA (page 8 and new Supplementary Fig. 1e).

4. In figure 2b, the SIRT4 loop looks like it packs against Tyr73 (via Pro200). Although the authors explore the possibility that the loop contributes to peptide binding, they do not check whether it affects binding of NAD⁺ to the enzyme. Given that this loop is a major structural difference between SIRT4 and other sirtuins, the authors should check whether deleting the loop alters apparent affinity for NAD⁺.

The Sirt4 loop indeed packs against Tyr73 (mentioned on page XX12), and we followed this thoughtful suggestion to compare the apparent NAD⁺ affinity of wildtype Sirt4 and a mutant lacking the Sirt4-loop. We included NAD⁺ titrations for both species that yielded values within the error of each other, showing that the loop alone does not significantly alter NAD⁺ binding, likely due to its conformational variability (page 12-13, 18).

5. The second distinguishing feature of the SIRT4 active site is the channel that branches off from the acyl-lysine binding tunnel and reaches the surface of the enzyme. This raises the possibility that a small molecule may stimulate the activity of SIRT4 (similar to SIRT6, where free fatty acids stimulate deacetylation activity JBC 2013). The authors should test whether adding free fatty acids, lipoic acid, or ketone bodies (i.e. beta-hydroxybutyrate or acetoacetate) stimulates SIRT4 activity against acetylated peptides, as well as HMG-ylated peptides.

We fully agree that the Sirt4-specific channel might act as a binding site for a regulatory metabolite and thank the referee for the insightful suggestion of candidate compounds. Testing them revealed an inhibitory effect of free lipoic acid, which fits quite well to our previous discussion of the channel as part of the binding site for longer acyl modifications and/or an externally accessible binding site for regulatory lipoylations. We included these results in the description (page 11) and discussion (page 19) of the Sirt4 channel as a potential regulatory binding site.

6. It is hard to reconcile the fact that HMG-ylation is removed far faster by SIRT5 than SIRT4 under the conditions tested in Fig. 1D, with the apparent specificity of SIRT4 for HMG-ylated peptides. If a full steady-state kinetic titration is done with SIRT5 and an HMG-ylated substrate, how does it compare to SIRT4-catalyzed de-HMG-ylation? In other words, does SIRT4 have higher k_{cat} for HMG-ylated peptides as compared to SIRT5? What is the overlap between SIRT4 and SIRT5 deacylating activities?

We measured a complete HMG substrate titration with Sirt5 under these conditions, and the resulting kinetic parameters match quite well with published (HPLC-based) values so that we included a reference to the published data instead of including the new ones. Both results show that Sirt5-dependent de-HMG-ylation is as efficient as our initial Sirt4 rate (from the coupled assay) and thus even exceeds the Sirt4 efficiency after correcting for HMG-induced NAD⁺ hydrolysis (see top #1). This is due to a higher k_{cat} for Sirt5, while K_M values for Sirt4 and Sirt5 are comparable. We included this information in the results (page 6-7). Although it cannot be fully generalized due to substrate sequence effects, we feel this result is in principle quite consistent with the fact that the Sirt5 active site is optimized for such dicarboxylate substrates, whereas Sirt4 appears to be more variable and likely dynamic in this active site region. We extended our treatment of acyl selectivities and isoform differences and overlaps in the results section (page 6-7) as well as the discussion (page 18) to clarify these points.

7. Accession codes need to be added from the PDB.

During revision of the manuscript we deposited crystal structures and diffraction data with the PDB and included the accession codes in a “Data Availability” statement at the end of the Methods section.

Minor Issues:

1. In Fig. 1C, it would be helpful to expand the low-substrate portion of the graph to show the differences in K_M between the different peptides.

Since the saturation level is equally important for the K_M value, we feel that expanding just the low-substrate part would not lead to a clearer illustration. Also, the only significant differences between K_M values are between acetyl-peptide and all other peptides, which we feel is well illustrated by Fig. 1C. Furthermore, all values are given with standard errors in Table 1, documenting the variation around the fit for each series, and we included an additional reference to Table 1 in the text (page 5) to make the reader aware of this additional documentation.

2. The authors should report $CC_{1/2}$ and CC^ statistics in Table 2.*

Since $CC_{1/2}$ and CC^* have a fixed and simple relationship ($CC^* = \sqrt{2 \cdot CC_{1/2} / (1 + CC_{1/2})}$) we feel that giving one of the two parameters provides all relevant information to the reader and followed the suggestion to add $CC_{1/2}$ values (as this is the appropriate parameter for judging information content of the diffraction data) in Table 2.

3. Do crystal contacts influence the conformation of the SIRT4-loop in the complex with ADP-ribose?

We included a statement that the Sirt4-loop is not involved in crystal contacts in our Sirt4/ADP-ribose complex structure (page 10).

4. In figure 3E/3F it would be helpful if the HMG-ylated peptide is displayed in a different color from the rest of the protein.

We followed the suggestion and changed the peptide color in 3e and 3f to “slate blue” to make it easier to distinguish protein and ligand. Also, we changed the protein color in 3e to “white blue” to make the color scheme consistent with 3f.

5. It would be interesting if the authors could speculate in the discussion about the biological significance of de-HMG-ylation by SIRT4 (i.e. conditions where ketone body synthesis increases such as fasting, connections to cholesterol synthesis via the mevalonate pathway, etc. and regulation by SIRT4).

We added a short comment on such conditions to the discussion section, with a reference to a recent paper from the Hirschey lab (see also referee #2, top #2) that treats the physiological relevance of protein HMG-ylation (page 17).

REVIEWERS' COMMENTS:**Reviewer #2 (Remarks to the Author):**

I believe that the authors have adequately addressed the concerns raised in my initial report. I recommend acceptance for publication.

Reviewer #3 (Remarks to the Author):

Given the new experiments and edits performed by Steegborn and coworkers, I recommend publication of this manuscript. I have one comment, which does not affect this recommendation.

With respect to HMGpeptide-stimulated NAD⁺ hydrolysis: As both HPLC - and mass spec-based assays can monitor turnover of NAD⁺ directly (by looking for the appearance of nicotinamide, disappearance of NAD⁺, or appearance of ADPribose versus acyl-ADPribose), this conclusion would be strengthened by this kind of analysis, rather than comparing turnover rates between two different kinds of assays.